**Resource**

# exRNA-eCLIP intersection analysis reveals a map of extracellular RNA binding proteins and associated RNAs across major human biofluids and carriers

## Graphical abstract

## Authors

Emily L. LaPlante, Alessandra Stürchler, Robert Fullem, ..., Eric Van Nostrand, Bogdan Mateescu, Aleksandar Milosavljevic

## Correspondence

mateescu@hifo.uzh.ch (B.M.), amilosav@bcm.edu (A.M.)

## In brief

By intersecting and deconvoluting extracellular RNAs (exRNAs) and eCLIP datasets, LaPlante and Stürchler et al. predict and validate novel extracellular RNA-binding proteins (exRBPs) associated with exRNA carriers in human biofluids and cell-conditioned medium. This work dramatically expands our current view on the role of exRBPs in cell-free RNA profiles.

## Highlights

- Intersecting eCLIP and exRNA dataset predicts extracellular RBPs in human biofluids

- Deconvolution of RNA bound by RBPs associates RBPs with exRNA carrier classes

- Generated RNA-RBP coverage map of more than 6,900 biofluid samples across whole human genome

- Differential release of extracellular RBPs and their isoforms across different cell types

LaPlante et al., 2023, Cell Genomics 3, 100303
May 10, 2023 © 2023 The Authors.

# Cell Genomics

CellPress

## Resource

# exRNA-eCLIP intersection analysis reveals a map of extracellular RNA binding proteins and associated RNAs across major human biofluids and carriers

Emily L. LaPlante,[1,19] Alessandra Stürchler,[2,3,19] Robert Fullem,[1,4] David Chen,[1] Anne C. Starner,[8] Emmanuel Esquivel,[1,4] Eric Alsop,[17] Andrew R. Jackson,[1] Ionita Ghiran,[5] Getulio Pereira,[5] Joel Rozowsky,[6] Justin Chang,[6] Mark B. Gerstein,[6] Roger P. Alexander,[7] Matthew E. Roth,[1] Jeffrey L. Franklin,[9,10] Robert J. Coffey,[9,10] Robert L. Raffai,[11,18] Isabelle M. Mansuy,[3] Stavros Stavrakis,[2] Andrew J. deMello,[2] Louise C. Laurent,[12] Yi-Ting Wang,[13] Chia-Feng Tsai,[13] Tao Liu,[13] Jennifer Jones,[14] Kendall Van Keuren-Jensen,[17] Eric Van Nostrand,[8,15] Bogdan Mateescu,[2,3,20,21,*] and Aleksandar Milosavljevic[1,4,16,20,21,22,*]

[1]Department of Molecular and Human Genetics, Baylor College of Medicine, Houston, TX 77030, USA
[2]Institute for Chemical and Bioengineering, ETH Zürich, 8093 Zürich, Switzerland
[3]Brain Research Institute, University of Zürich, 8057 Zürich, Switzerland
[4]Program in Quantitative and Computational Biosciences, Baylor College of Medicine, Houston, TX 77030, USA
[5]Department of Anesthesia, Beth Israel Deaconess Medical Center, Harvard Medical School, Boston, MA 02215, USA
[6]Department of Molecular Biophysics and Biochemistry, Yale University, New Haven, CT 06520, USA
[7]Extracellular RNA Communication Consortium, Phoenix, AZ 85254, USA
[8]Verna & Marrs McLean Department of Biochemistry & Molecular Biology, Baylor College of Medicine, Houston, TX 76706, USA
[9]Department of Medicine, Vanderbilt University Medical Center, Nashville, TN 37232, USA
[10]Department of Cell and Developmental Biology, Vanderbilt University School of Medicine, Nashville, TN 37235, USA
[11]Department of Veterans Affairs, Surgical Service (112G), San Francisco VA Medical Center, San Francisco, CA 94121, USA
[12]Department of Obstetrics, Gynecology, and Reproductive Sciences and Sanford Consortium for Regenerative Medicine, University of California, San Diego, La Jolla, CA 92093, USA
[13]Biological Sciences Division, Pacific Northwest National Laboratory, Richland, WA 99354, USA
[14]Laboratory of Pathology Center for Cancer Research, National Cancer Institute, Bethesda, MD 20892, USA
[15]Therapeutic Innovation Center, Baylor College of Medicine, Houston, TX 77030, USA
[16]Dan L. Duncan Comprehensive Cancer Center, Baylor College of Medicine, Houston, TX 77030, USA
[17]Neurogenomics Division, TGen, Phoenix, AZ 85004, USA
[18]Division of Endovascular and Vascular Surgery, Department of Surgery, University of California, San Francisco, San Francisco, CA 94143, USA
[19]These authors contributed equally
[20]These authors contributed equally
[21]Senior author
[22]Lead contact
*Correspondence: mateescu@hifo.uzh.ch (B.M.), amilosav@bcm.edu (A.M.)

## SUMMARY

Although the role of RNA binding proteins (RBPs) in extracellular RNA (exRNA) biology is well established, their exRNA cargo and distribution across biofluids are largely unknown. To address this gap, we extend the exRNA Atlas resource by mapping exRNAs carried by extracellular RBPs (exRBPs). This map was developed through an integrative analysis of ENCODE enhanced crosslinking and immunoprecipitation (eCLIP) data (150 RBPs) and human exRNA profiles (6,930 samples). Computational analysis and experimental validation identified exRBPs in plasma, serum, saliva, urine, cerebrospinal fluid, and cell-culture-conditioned medium. exRBPs carry exRNA transcripts from small non-coding RNA biotypes, including microRNA (miRNA), piRNA, tRNA, small nuclear RNA (snRNA), small nucleolar RNA (snoRNA), Y RNA, and lncRNA, as well as protein-coding mRNA fragments. Computational deconvolution of exRBP RNA cargo reveals associations of exRBPs with extracellular vesicles, lipoproteins, and ribonucleoproteins across human biofluids. Overall, we mapped the distribution of exRBPs across human biofluids, presenting a resource for the community.

## INTRODUCTION

Extracellular RNAs (exRNAs) are RNAs that exist in biofluids outside of cells and can be associated with a variety of carriers that includes vesicles, lipoproteins, and free ribonucleoproteins (RNPs; RNA-binding proteins plus the RNA bound to them).[1] Because exRNAs offer insights into the cell from which they originate, they provide a bounty of information that can be leveraged

for liquid biopsy detection and tracking of normal and disease processes. Previously, in an effort to compare data across studies, which is often confounded by lab-specific variables, deconvolution revealed 6 major cargo types (CTs) that could be related to high- and low-density vesicles (HDVs and LDVs, respectively), lipoprotein particles (LPPs), and RNPs.[2] However, specific RNA-binding proteins (RBPs) carrying exRNA (exRBPs) were not identified, with the exception of AGO2. AGO2 was associated with CT3A and CT3B CTs by sequencing AGO2 pull-downs followed by correlating the RNAs in the AGO2 profile with RNAs in the CT3A and CT3B profiles. In contrast, despite CT3C also likely being associated with an RNP carrier type, it showed little correlation with any reference CT tested, opening the question about the specific exRNA carrier involved. Given that there are over 2,000 RBPs in human cells,[3] and three of six CTs (CT3A, CT3B, and CT3C) are related to RNPs, we concluded that the extent to which RBPs serve as exRNA carriers was worth exploring. This major gap in knowledge limited our understanding of the role of RBPs in exRNA biology and their potential as liquid biopsy biomarkers that bind and export exRNA fragments of human genes into accessible human biofluids.

A major role of RBPs as exRNA carriers would be expected based on their role in the biogenesis, localization, loading, and transport of exRNA.[4–10] Generally, exRNA research has focused largely on vesicular exRNAs because of the difficulties associated with identifying the diversity of RBPs (∼2,000) that can potentially serve as carriers of exRNA. While mass spectrometry experiments led to the discovery of RNP structures like exomeres and supermeres,[11,12] the specific RBPs involved in these structures and their exRNA cargo remain unknown. New opportunities to address these questions became possible following the release of enhanced crosslinking and immunoprecipitation (eCLIP) profiles as part of the ENCODE project. These eCLIP profiles allowed precise detection of binding sites of 150 RBPs across the human genome.[13] While these results reflect intracellular RBP-RNA interactions, we hypothesized that similar interactions may be observable extracellularly as well. By combining the cellular eCLIP profiles from ENCODE with human small RNA sequencing (RNA-seq) from different biofluids in the exRNA Atlas, it became possible to determine which RBPs play a role as exRNA carriers.

The most direct combined analysis approach to identify an exRBP is to ask whether exRNA profiles in the Atlas contain a significant amount of exRNA fragments in the regions predicted by eCLIP to bind with the RBP. While informative, this direct method alone has limited power to establish an association of a specific exRNA fragment with a specific RBP in extracellular space. Moreover, we assumed that the power to detect a footprint at the level of an RBP could be enhanced because the relative amounts of any RBPs and their cargo vary from sample to sample because of biological and experimental variables. Specifically, we reasoned that, when the expression levels of exRNA fragments carrying the binding signature of the same RBP covary across diverse conditions, that "correlation footprint" indicates a specific interaction between that RBP and those exRNA fragments. The interaction might either be direct, as in an extracellular RNP complex, or indirect, resulting from a role of the RBP in the export of the exRNAs into extracellular space. By combining evidence

from individual fragments, we predicted that we could obtain evidence about any possible role of the RBP as an exRNA carrier. We applied this "correlation footprint" strategy to develop a map of candidate exRBPs and their exRNA cargo.

Here, we present analyses of this exRBP map resource as well as its experimental validation. Extending correlation analysis to specific RNA biotypes, we identified biotypes carried by the exRBPs, placed the exRBPs within specific human biofluids, and associated them with different classes of extracellular vesicles (EVs), lipoproteins, and RNP particles. We also identified exRBPs within different classes of EVs produced by red blood cells (RBCs). Cognizant of the vast opportunities for further exploration of exRBP biology and their potential as disease biomarkers, we offer an exRNA Atlas resource with exRBP coverage information for all 150 eCLIP-profiled RBPs available for all 6,930 human small RNA-seq datasets.

## RESULTS

### Extending the exRNA Atlas resource with RBP exRNA profiling information

To identify candidate exRNA fragments carried by RBPs, we analyzed ENCODE eCLIP data for 150 RBPs (as described in STAR Methods). Guided by the hypothesis that RNA bound inside cells would also be found in the biofluids and are identifiable in exRNA profiles, we intersected eCLIP RBP binding sites from ENCODE with human small RNA-seq data derived from cerebrospinal fluid (CSF), plasma, saliva, serum, and urine from the exRNA Atlas (Figures 1A–1C).

The intersection between eCLIP, RBPs, and exRNAs required base-pair-level resolution of read coverage in the human small RNA-seq profiles and an ability to identify specified regions captured in the ENCODE eCLIP BED files. Because of the large number of samples in the Atlas and the scalability required to accommodate even larger numbers of samples in the future, we needed a compact file format for the read coverage and the intersect files. We thus repurposed bedGraphs, which were originally designed as a compact annotation track format to visualize genome-wide tracks at single-base-pair resolution. In addition, bedGraph files are compatible with BEDTools, making file manipulation efficient and accessible for downstream computations.

Briefly, we aligned human small RNA-seq profiles from the exRNA Atlas to hg19 to create BAM files that were converted into bedGraphs (Figure 1D). Hg19 was selected to maintain compatibility with legacy data in the exRNA Atlas and avoid realignment of the entire exRNA Atlas. The bedGraphs were then intersected with the eCLIP-derived RBP binding sites using the BEDTools map function. We note that because the typical exRNA and eCLIP fragments were much shorter than sequencing read length, coverage of each fragment could be measured as the count of reads mapped. We therefore counted the number of unique reads that intersected with each RBP binding site in every individual sample (Figure S1A). Next, files were collapsed in two ways: (1) per individual RBP and study-specific dataset, providing read counts at each binding site for all the samples in the dataset, and (2) per individual sample, providing read counts at each binding site for all RBPs. We found that 61% of RBP binding sites had at least 5 unique reads in at least one

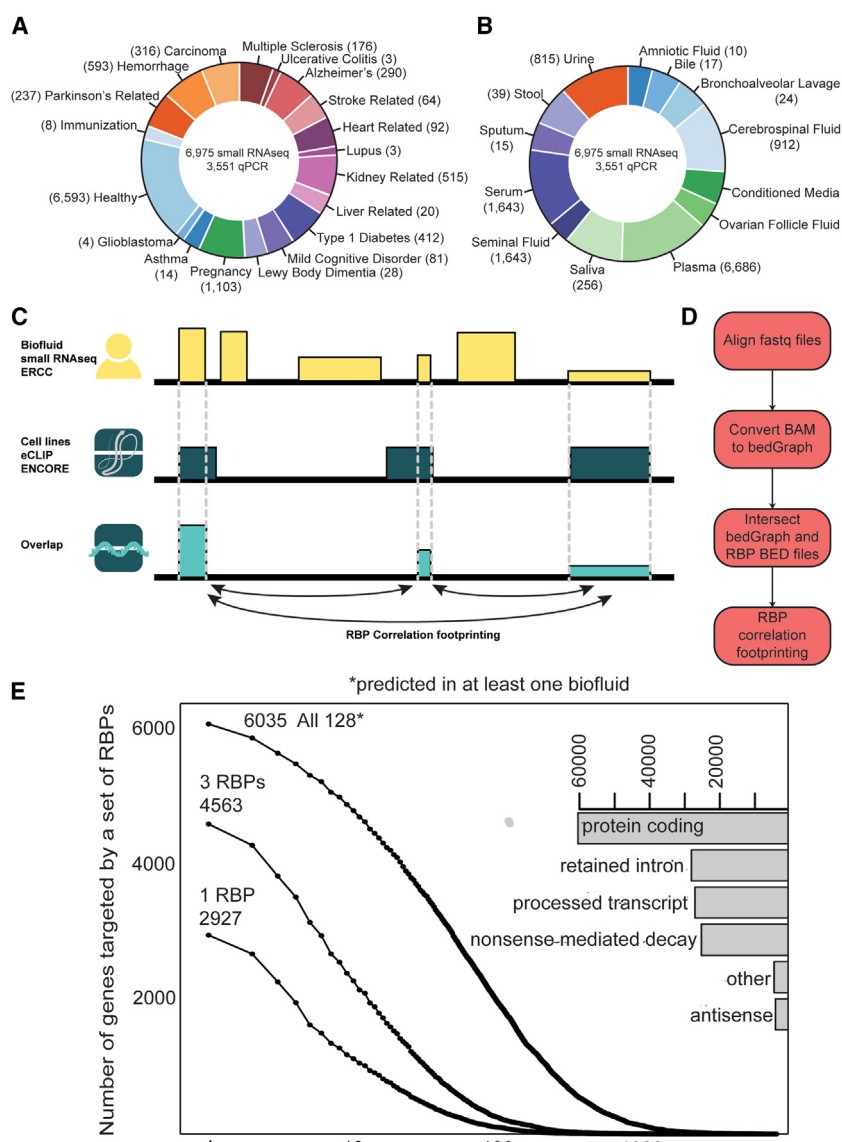

**Figure 1. Intersection of the ERCC exRNA Atlas and ENCORE RBP eCLIP profiles identifies exRBPs and their exRNA cargo**

(A) Distribution of biosamples in the exRNA Atlas by condition. The size of each slice represents the log-transformed number of samples for a given condition.

(B) Distribution of biosamples in the exRNA Atlas by biofluid. The size of each slice represents the log-transformed number of samples for a given biofluid.

(C) Schematic of overlap detection between ERCC exRNA Atlas profiles and loci corresponding to transcribed fragments bound to a specific RBP, as identified by ENCORE eCLIP.

(D) Process of intersecting profiles in the ERCC exRNA Atlas with regions identified by ENCORE eCLIP, followed by Pearson correlation footprinting to identify putative exRBPs and their exRNA cargo.

(E) Count of human protein-coding genes (y axis) that are sampled by different combinations of RBPs (1, 3, 128 distinct RBPs) at minimal per-gene counts of RBP-bound fragments (x axis). The three curves correspond to RBP combinations of different sizes (1, 3, 128). For the "1 RBP" curve, the y value for each point is calculated for the RBP that covers the largest number of genes for a given per-gene coverage threshold (x value); thus, each point may correspond to a different RBP. Similarly, the y value for each point along the "3 RBP" curve is calculated for the triplet of RBPs that covers the largest number of genes; thus, each point may correspond to a different combination of RBPs. Horizontal bars in the top right of the panel indicate counts of genomic annotations for all RNA fragments associated with all 128 RBPs; note that a single fragment may have more than one annotation.

exRNA sample and were able to identify RBPs in human biofluids where a higher-than-random coverage was present, allowing us to collect evidence of the presence of specific RNA-RBP interactions.

## Correlation analysis reveals footprints of 34 RBPs in cell-line-conditioned medium and human plasma

We next performed a Pearson correlation analysis between loci bound by a single RBP to identify "footprints" of the RBP in conditioned cell- line culture medium and from human plasma. This analysis was motivated by results from our previous Atlas analysis that suggest large sample-to-sample variation in the relative abundance of exRNA carriers.[2] Under that assumption, stronger-than-expected covariation of exRNA at an RBPs target locus determined by Kolmogorov-Smirnov test would represent a "footprint" suggesting that the RBP carries the exRNA.

Correlation analysis compiled evidence of the presence of each locus in the extracellular space. Specifically, for a single index RBP, every locus was correlated with all other loci within that same RBP and the distribution of those Pearson correlations generated (Figure S1B). Subsequently, as a negative control, a locus with a similar read coverage but belonging to a different RBP was correlated with the loci from the index RBP and the distribution of those Pearson correlations generated (Figure S1B). A Kolmogorov-Smirnov test was then applied to compare these two Pearson correlation distributions to determine whether the locus being tested was significantly more correlated with the other loci of its RBP than expected by chance (Figure S1C).

Correlation analyses for all individual loci of an RBP were next combined to provide evidence of the RBP as a whole. Specifically, for each RBP, the distribution of p values for the Kolmogorov-Smirnov tests for each RBP-bound locus were compared with p value distributions generated from sets of random loci with coverage similar to the original loci (Figure S1C). An RBP "footprint" was declared whenever the distribution of

## A

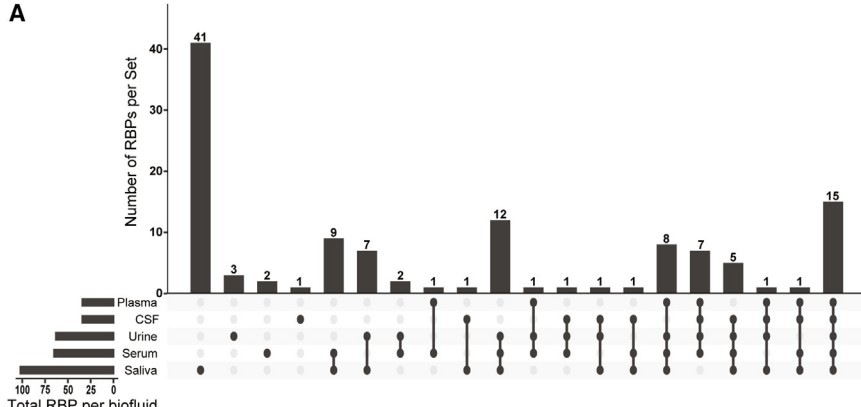

Figure 2. Many exRBPs show correlational exRNA footprints and consistent biotype bias across different biofluids

(A) Upset plot for distribution of exRBPs across biofluids.

(B) exRBPs show consistent cargo bias for Y RNAs, tRNAs, snRNAs, snoRNAs, piRNAs, and miRNAs across biofluids where they are detected.

## B

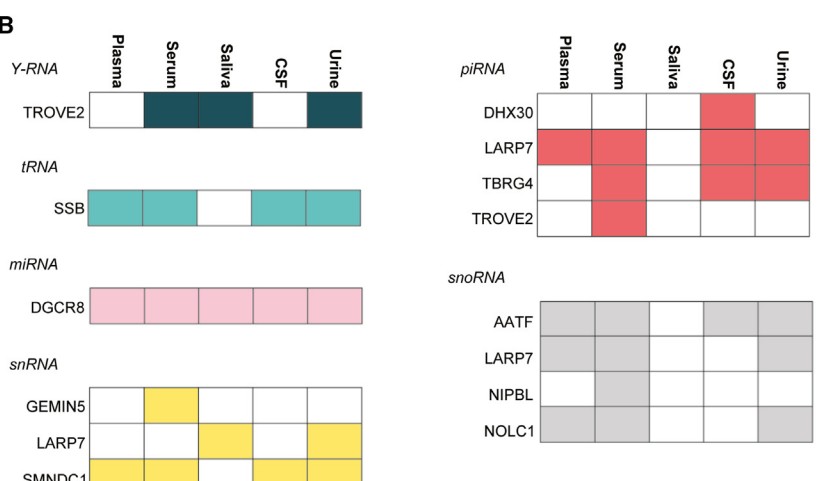

### RBPs carry exRNA representing at least 30% of human protein-coding genes

We next expanded our correlation footprinting from just cell-line-conditioned medium and plasma to all biofluids well represented in the exRNA Atlas. We focused on plasma, serum, saliva, CSF, and urine because these five biofluids had at least one Atlas study with more than 40 samples. To decrease variability, we focused only on healthy samples from each of the large (>40 samples) studies involving the five biofluids (plasma = 789, saliva = 158, serum = 218, urine = 300, CSF = 248). We then performed a correlation footprint analysis for the same set of RBPs in the additional 4 biofluids and found targets for 128 of 150 RBPs to be present in at least one biofluid (Figures 2A and S2A). The footprints of common small RNA-processing RBPs, such as DGCR8 (miRNA)[14] and SSB (tRNA)[15] were detected in all or most biofluids, while some RBPs were unique to certain biofluids (Figure S2A). Similarly, RBP binding loci were found across all biofluids or were expressed in a biofluid-specific manner (Figures S2B and S2C). 30% of known human protein-coding genes were represented by at least one exRNA fragment bound to at least one of the 150 RBPs across all five biofluids. This coverage percentage was stable even when considering only the 128 RBPs with "correlation footprint" evidence. As expected, the number of represented genes decreased as we increased the required number of distinct fragments per gene (Figure 1E).

### RBPs show exRNA biotype-specific bias

Based on their biology, some RBPs are expected to show bias for specific small RNA biotypes, including miRNA, piRNA, tRNA, small nuclear RNA (snRNA), small nucleolar RNA (snoRNA), and Y RNA. To test and thereby indirectly validate our method, we focused our "correlation footprinting" analysis on subsets of RBPs that fall within specific RNA biotypes. For each RBP, the loci within biotypes were correlated and the p values compared with p values generated from random sets of loci. The analysis revealed substantial bias for some of the RBPs (Figure 2B). For example, the protein SSB, which processes tRNAs,[15] shows high Pearson correlation over tRNAs

p values for true loci was significantly shifted toward lower values compared with the control distributions determined by Kolmogorov-Smirnov tests. We began by testing the RBP footprints in cell-line-conditioned medium (conceptually closest to the original cell line eCLIP data) and then human plasma (the largest set of samples). We found 34 RBPs that were significantly correlated (Kolmogorov-Smirnov test with a Bonferroni adjusted p < 0.05; Figure S1D) in both sets, suggesting their role in exRNA transport and/or stabilization in cell-conditioned medium and plasma (Figure S1D).

We next asked whether specific categories of RBP functions were enriched among those carrying exRNA vs. all of the 150 that were originally profiled using eCLIP and classified by ENCODE. The following five classes were determined to be enriched among those carrying exRNA relative to the full set of RBPs: spliceosome RNAs, microRNA (miRNA) processing, RNA localization, viral RNA regulators, and other (Figure S1E). We also looked at the annotations of RNA fragments bound to the RBPs and found a distribution similar to other eCLIP-profiled RBPs (Figures 1E and S1F). Taken together, our results indicate that the RBPs selected by our footprinting pipeline do not show strong bias toward specific genic regions or RBP functions compared with other RBPs profiled by eCLIP.

bound by SSB in all biofluids except saliva, while DGCR8, a miRNA-processing RBP,[14] is detected in all biofluids (Figure 2B). Interestingly, GEMIN5, which carries pre-snRNAs,[16] is only detected in serum, suggesting the presence of non-complete spliceosomes. In contrast, targets of SMNDC1 (SPF30/SMNrp), which is found in complex A and B spliceosomes,[17] are found in a wider variety of biofluids, suggesting that complete spliceosomes may be present in these biofluids. Taken together, these analyses confirm expected biotype bias while revealing some intriguing differences in RNA cargo for the same RBP across different biofluids.

Finally, we tested the assumption that restriction of correlation footprinting to RBP-unique loci (based on eCLIP) vs. any RBP-associated loci improved the performance of our method. We focused our analysis on CSF. Restriction to unique loci resulted in higher RBP numbers predicted (Figure S2D). Moreover, all 9 RBPs that were predicted from unique loci and not predicted based on all (including non-unique) loci were experimentally validated by western blotting (Figure S2D). The downside of removing sites bound by multiple RBPs was that some RBPs were no longer prioritized, and, thus, they were not tested in all biofluids.

### Experimental validation of computationally predicted exRBPs

We next proceeded to validate 34 computationally predicted exRBPs (all 34 predicted in cell medium and plasma) by experimentally testing their presence in different cell-conditioned medium (CM). We identified 20 antibodies for a subset of these RBPs and performed western blots in cell lysate and CM isolated from three distinct *in vitro* human model systems: embryonic kidney 293T cells (293T), human mesenchymal stem cells (hMSC), and DiFi cells.[18] 293T is a widely used cancer cell line that grows at high density. hMSCs derived from human bone marrow are a primary cell line with multipotent cell characteristics and a model for studies of EV biogenesis and function. DiFi colorectal cancer cells are cultured in hollow-fiber bioreactors, allowing large-scale production of CM; DiFi cells are also the reference cell model for the exRNA Communication Consortium (ERCC2).[19]

293T and hMSCs were cultured in serum-free medium for 48 h. CM was collected, spun at 1,000 × g to remove cells and large debris, filtered using a 0.45-μM pore size syringe filter, and then concentrated with an ultrafiltration cartridge (Figure 3A). Cells from the same culture dishes were also collected in parallel. Protein extracts from cellular and CM were prepared from three independent experiments and used as input for western blot experiments to calculate the corresponding extracellular/intracellular ratios (Figures 3B and 3C). Cellular and CM protein extract from DiFi cells grown in a hollow-fiber bioreactor were generated and tested by western blot in another laboratory as a parallel independent validation experiment (Figure S3A).

In the 293T experiment, 78% (14 of 18) of the tested RBPs were consistently detected in CM protein extract replicates. Based on calculation of the extracellular/intracellular ratios, 50% (9 of 18) of the RBPs were overrepresented in CM rather than cell lysate among all tested RBPs, showing a range of enrichment varying from 1.4 to almost 700. Testing the same panel of antibodies in hMSCs revealed a reduced number of

detectable RBPs (13 of 18), with lack of signal for AQR, EFTUD2, LIN28B, ZNF622, and PPIG. These results largely agree with the transcriptomics and proteomics literature in bone-marrow-derived hMSCs[20] that indicates very low or even absent expression of those genes. For the quantifiable RBP candidates, 61.5% (8 of 13) were detected in CM, and 31% (4 of 13) had a higher relative abundance in the extracellular space than in cells, showing a range of enrichment from 2.1–6.3 (Figures 3B, 3C, and S3B). Regarding DiFi cells, 57% (8 of 14) of the detectable RBPs were found in the CM (Figures S3A and S3B).

Comparing the western blot panels from all cell types, we observed consistency with 56% (10 of 18) of the RBPs showing matching results for the cell types tested (we only considered RBPs present in at least two panels). From these 10 RBPs, we concluded that 8 were consistently detected in the extracellular space—i.e., AQR, EFTUD2, FTO, GRWD1, ILF3, SF3A3, SSB, and XPO5 (Figures 3B, 3C, S3A, and S3B). Concerning the remaining RBP candidates studied in those panels, we found that DGCR8, NCBP2, PRPF8, RPS3, SF3B4, and ZNF622 showed different outcomes depending on the cell type tested. This result suggests that cell origin specifies the mechanism behind the release or stabilization of RBPs in the extracellular space.

To increase the number of validations, 5 additional RBPs computationally predicted to be present in CM but not in plasma were validated in 293T, and their extracellular/intracellular ratios were calculated (Figures S3C and S3D). We had a high prediction success with 4 out of 5 (80%) RBPs detected in the extracellular space and with TROVE2 and YBX3 showing an enrichment in CM compared with cell lysate (Figures S3C and S3D).

Interestingly, we noted that, in some cases, the protein signal in CM appeared to be isoform specific. For example, ILF3 exhibited expression of two main isoforms of molecular masses of 90 and 110 kDa and a relative enrichment of the smaller isoform in the CM mainly in 293T or only in hMSCs (Figure 3B). These two ILF3 isoforms are described in the literature as NF90 and NF110, and while the 110-kDa isoform is primarily nuclear, the 90-kD isoform is also known to shuttle to the cytoplasm.[21] Similarly, we observed that QKI showed enrichment of a lower-size isoform in the CM, while the more abundant and higher-size variant is depleted (Figure S3C). QKI expresses at least three isoforms (QKI-5, QKI-6, QKI-7) through alternative splicing mechanisms.[22] The longest and more abundant isoform (QKI-5) is predominantly localized in the nucleus, while the smaller isoforms are either distributed between the nucleus and cytoplasm (QKI-6) or primarily in the cytoplasm (QKI-7).[22] Based on these observations, it is likely that the lower-molecular-weight ILF3 and QKI isoforms enriched in 293T CM correspond to their cytoplasmic form and that those RBPs may be involved in subcellular-specific export. Regarding DGCR8 in 293T, we could detect a lower-molecular-weight isoform enriched in CM reported previously to be differentially expressed across colon cancer cell lines,[23] pointing to another example of release or stabilization of RBP isoforms in the extracellular space (Figure 3B).

Importantly, some RBPs could only be detected in cells (DDX3X, BCLAF1, and TRA2A), while others could be found enriched in CM (AQR, EFTUD2, and GRWD1), as shown in the 293T

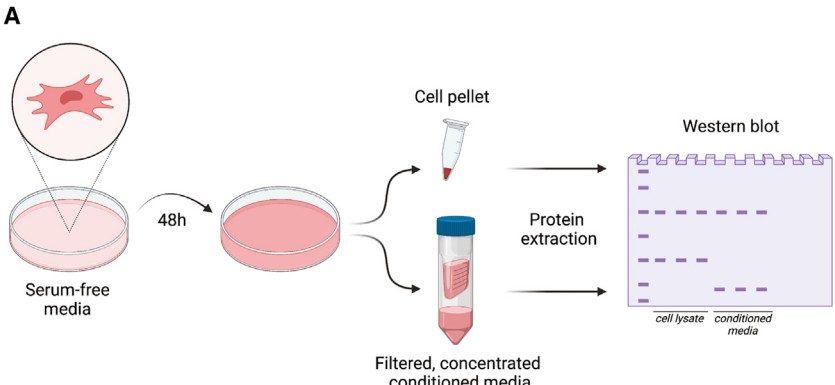

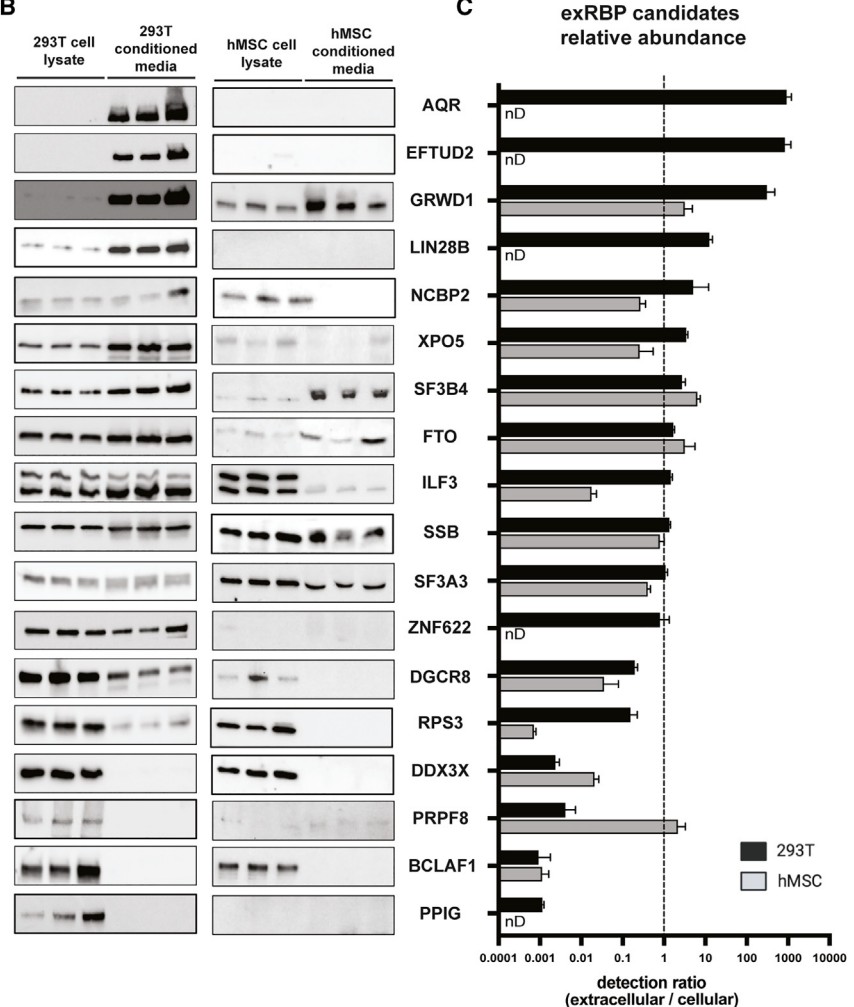

**Figure 3. blots validate and quantify exRBPs in cell-conditioned media**

(A) Diagram of serum-free cell culture for protein extraction from paired cell lysate and concentrated conditioned media and following RBP detection by western blot. Created with BioRender.

(B) Western blots performed in triplicate on 293T and hMSC cell lysate and conditioned media to quantitate ratios in cell-conditioned media vs. cell lysate.

(C) Ratio of detected signal between conditioned media and corresponding cell lysate blots for 293T and hMSCs. A value greater than 1 indicates stronger detection of the RBP in the extracellular space, while a value below 1 shows stronger detection intracellularly. "nD" indicates lack of intracellular detection in hMSCs. Data are represented as mean +SD.

turnover within cells and stability in the CM. Detecting RBPs only in cell lysate and not in CM, on the other hand, indicates either the existence of a very tightly controlled mechanism of protein export by the cells or the presence of a harsh environment for some RBPs outside of the cell that may result in rapid degradation.

To validate RBP detection in plasma, we leveraged the Plasma Proteome Database (PPD)[24,25] as an alternative and independent discovery tool. 24 of 34 RBPs had been detected previously in plasma via mass spectrometry. Overall, the extracellular presence of 27 of 34 (79%) computationally predicted RBPs could be corroborated either by our western blots of CM or by previous mass spectrometry of plasma, and the extracellular presence of 12 of the RBPs could be corroborated by both types of evidence (Figure S3B).

## RBP exRNA cargo associates with specific vesicular and non-vesicular exRNA carriers

We next asked whether the exRBPs could be associated with specific vesicular and non-vesicular carriers of exRNA. The original analysis of the exRNA Atlas resource revealed distinct profiles (CTs) associated with EVs, lipoproteins, and free RNPs.[2] We took the proportions of each carrier estimated via deconvolution from the original exRNA Atlas and estimated the expression profile of the RBP binding loci of each RBP in each CT. To assign RBPs to a specific CT, we created an RBP-CT enrichment score. This score was the ratio of the average number of RBP-bound reads that fell into a specific CT divided by the average number of RBP reads across all CTs. We then visualized the RBP-CT enrichment scores and

panel (Figures 3B and S3A–S3D). Of note, 293T and hMSC protein extracts were loaded at a ratio so that a majority of RBPs could be detected at similar levels in the cell and CM. This may explain why some RBPs would be below the level of detection in the cell extract. Alternatively, cases where we detected an RBP only in the extracellular space could be explained by high

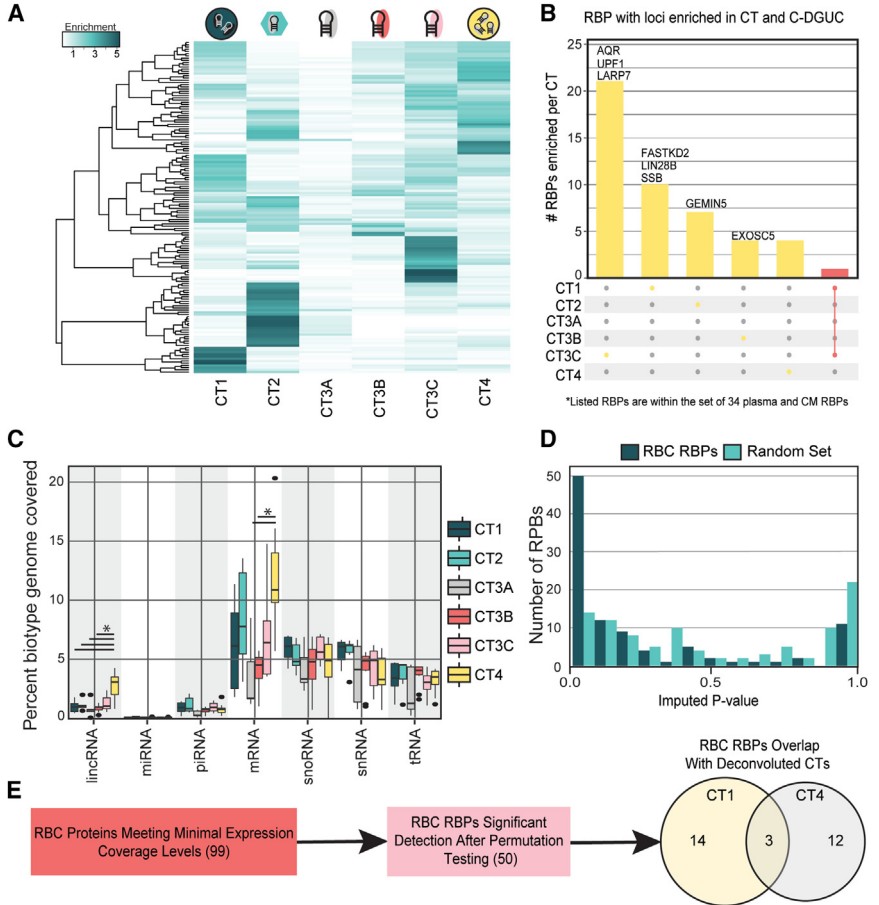

**Figure 4. exRBPs associate with specific vesicular and non-vesicular exRNA carriers**

(A) Enrichment of exRBP-specific exRNA cargo within specific vesicular and non-vesicular carrier cargo types (CTs).

(B) Upset plot depicting the number of RBPs that show consistent enrichment for specific CTs by multiple methods.

(C) Enrichments for specific exRBP biotype cargo by CT tested by Kruskal-Wallis test. Whiskers extend to min and max, and outliers are greater than 1.5 interquartile range (IQR). *p < 0.05.

(D) Distributions of exRBPs p values of coefficient of variance tests across loci vs. permuted locus coefficient of variance.

(E) Computational assignment of candidate RBC exRBPs to CT1 and CT4 CTs.

clustered them accordingly to identify sets of RBPs that were present in a similar set of CTs (Figures 4A and S4A).

Interestingly, 21 of our set of 34 cell line- and plasma-associated RBPs were enriched for CT3C, and 54 RBPs were enriched for CT3C among the entire set of 150 RBPs (Figures 4B, S4B, and S4C; Table S1). CT3C stood out as not associated with any known carrier in our original analysis.[2] Unlike other RNP-associated CTs (CT3A and CT3B), it showed no Pearson correlation with RNA-seq profiles obtained from AGO2 pull-downs.[2] This therefore suggests that CT3C encompasses specific exRNA carriers.

To further corroborate this result using independent experimental data, we used the RNA-seq profiling data derived from plasma processed with cushioned density gradient ultracentrifugation (C-DGUC) fractions that were originally used to identify different exRNA carrier types.[2,26] Fractions from C-DGUC originally correlated with the different carrier classes (fractions 1–3 with CT4 [HDVs], fractions 4–7 with CT1 [LDVs], and fractions 9–12 with CT2 [LPPs]), while unfractionated serum and plasma correlated with CT3 (A, B, and C).[2] Using these reference C-DGUC RNA-seq profiles, we created base-pair-level read density maps in the form of bedGraphs and intersected them with the eCLIP profiles of the 150 RBPs to create profiles of each RBP within individual C-DGUC fractions. For each RBP and each fraction, we then examined coverage of the RNA loci

associated with the RBP within the fraction. In cases with high coverage within one fraction relative to others, we used this as evidence for the presence of the RBP in that fraction. As an additional indicator, we examined sample-to-sample correlations within the C-DGUC reference classes between the amount of RBP cargo and the amount of computationally deconvoluted fraction-specific RNA CT. Our analysis focused on RBPs that could be concordantly assigned to distinct CTs by both methods (Figure 4B; Table S2). Our analysis revealed significantly more lincRNA in CT4 than in all CTs examined except CT3C (Wilcoxon rank-sum test; CT4 vs. CT1, p = 0.01; CT2, p = 0.04; CT3A, p = 0.02; CT3B, p = 0.001). We observed significantly more mRNA in CT4 than CT3B (Wilcoxon rank-sum test, p = 0.00005) (Figure 4C). Although miRNAs are detected in very low proportions across the exRBP-associated loci, many miRNA-processing proteins are absent from the analyzed set of 150 proteins, most notably AGO2, which carries diverse miRNAs extracellularly. The exRBPs in our set also do not overlap with many pri-miRNAs compared with what is detected in a more comprehensive analysis of exRNA (Figure S4D).

To validate RBP presence in EVs (CT1 and CT4), we asked whether the exRNA cargo associated with RBPs can be found in RBC-derived EVs. For this purpose we isolated four EV-enriched samples derived from RBCs and profiled them by small RNA-seq. We focused our analysis on the 99 (of 150) RBPs with at least 15 loci remaining after filtering for loci with greater than 0 reads in at least two of the four samples. We compared the distribution of variance in locus coverage across the 4 RBC samples against the variance distribution of re-sampled loci and repeated the analysis over randomized locus sets for each RBP. We found that 50 of 99 RBPs showed significantly lower variance while only 15 of 99 randomized locus sets showed significantly lower variance (two-sample Kolmogorov-Smirnov test, p = 9.257 × 10$^{-7}$), suggesting an approximate

30% false discovery rate for the 50 detected RBPs (Figure 4D). Furthermore, compared with the RBP-CT deconvolution Pearson correlations, 58% (29 of 50) of the RBC EV-associated RBPs were associated with vesicular fractions CT1 and CT4 (Figure 4E). These findings reinforce the hypothesis of the presence of a large number of RBPs in RBC-derived EVs. Moreover, these results suggest that our exRBP data resource, in conjunction with independent profiling experiments (such as RNA-seq performed on RBC EVs), may help to accurately identify the cellular source of exRBPs.

### exRBP extension of the exRNA Atlas resource

To enable further exploration of exRBP biology and exRBP biomarker discovery by the community, we extended the exRNA Atlas resource by including exRBP information. All generated bedGraph and intersect files were compressed, stored, and made accessible in the post-processed and core results files via the exRNA Atlas website and API, resulting in 2,107,242 files available via the Atlas (STAR Methods). As new samples are submitted to the exRNA Atlas, the bedGraph and intersect files will be made available along with exceRpt processing results. The new resource expands the original exRNA regions of interest from tens of thousands of genes to ~1.4 million specific regions (most being specific RBP-associated fragments of long transcripts). This will provide the scientific community with a way to easily access them as well as provide opportunities for users to investigate regions relevant for answering their research questions. Users can also download intersect files for all of the 150 RBPs currently available in ENCODE and investigate in a sample-, biofluid-, and condition-specific manner or use bedGraphs for all 6,930 samples from the atlas and explore specific regions of interest.

All information and tools are publicly available at GitHub (https://genboree.org/theCommons/projects/exrna-mads/wiki/ExRNA_Atlas) and Zenodo (https://doi.org/10.5281/zenodo.7706896). Beyond the information for pre-defined bedGraphs (e.g., those from ENCODE eCLIP experiments), to promote analysis of any exRNA profiles over any other regions of interest, we created a data slicing tool where users can provide their own regions of interest and select samples from the Atlas to intersect. This tool is available for download in the "bedGraph" section.

Finally, we created annotation files for the eCLIP regions. Annotation files note whether reads were found in exRNA samples split on biofluid and library preparation kit. The presence of RBPs was determined by coverage of 1 or 5 unique reads at each genome location when the region was present in at least one sample in the set at or above the coverage cutoff. These files can be downloaded and viewed in the UCSC genome browser. This tool is available for download in the "bedGraph" section. For those interested in viewing how the RBP binding sites are expressed across many samples and datasets, we added the RBP annotations in the exRNA Explorer Tool, which can be accessed via the exRNA Atlas (https://www.exrna-atlas.org) interface or the "exRBP" section of the page linked above.

In summary, we provide a single point of access for exRBP resources, including downloadable base-pair-level coverage maps, tools to interrogate regions of interest in any set of samples, and visualization-ready genome browser tracks with detected exRNA regions.

## DISCUSSION

Our results shed light on the role of RBPs as exRNA carriers across human biofluids. The exRBP map construction leveraged the power of large, publicly available datasets to study existing data in new ways to produce more data, tools, and resources available to the public to further empower research within an integrated experimental and computational research paradigm.

The 128 RBPs we detected in at least one biofluid using a "correlation footprinting" strategy were found to bind 30% of coding genes and cover ~1% of the genome in our exRNA Atlas samples. These results lay the foundation for a class of biomarkers, where exRNA fragments bound to exRBPs increase the number of detectable markers in biofluid. Moreover, the dual (RNA/protein) specificity of exRNA-exRBP complexes provides potentially high accuracy for monitoring pathological processes and potentially high positive predictive value required for early disease detection.

The exRBPs predicted by correlation footprinting were validated experimentally at a nearly 80% rate in plasma (27 of 34) and cell culture CM (31 of 39), suggesting high specificity of our computational method. However, a number of analytical design decisions may have affected sensitivity and validation rate. First, we made the decision to exclude loci binding multiple RBPs (based on eCLIP) from correlation footprinting and focus on loci that are more likely to be RBP specific. This choice was expected to achieve a better experimental validation rate with only moderate cost in terms of loss of sensitivity. The restriction of correlation footprinting to unique loci had, in fact, mixed effect on sensitivity with improved sensitivity for RBPs associated with many unique loci. However, some RBPs had no regions that were unique to them and therefore could not be tested. Second, a number of RBPs are core processing, splicing, or loading RBPs, which means that the RNAs they bind could be released after being processed while they are not in complex with the RBP. In this case, Pearson correlation of RNAs bound by an RBP does not indicate the RBP's presence, simply that it has a role in processing RNAs for export. The third factor is technical and biological variation, as evidenced by discrepancies between RBPs detected in 293T, hMSC, and DiFi cell lysate (Figures 3B, S3A, and S3C). Of note, CM experimental procedures were shared for 293T and hMSCs, with a different protocol for DiFi (different culture medium, 2D culture for 293T/hMSCs, and 3D culture for DiFi cells, which have a distinct protein extraction method). Moreover, the samples were prepared and experiments were carried out by two independent research groups (one laboratory for 293T/hMSCs and the other for DiFi cells). However, despite these differences, several RBPs were found in CM of all three cell lines, suggesting that their detection in the extracellular space is largely independent of cell types, experimental procedures, or laboratories.

Our initial hypothesis was that RNA sequences bound by specific RBPs within cells (as monitored by eCLIP data) would protect the RNA fragments detected in biofluids (as monitored by exRNA sequencing data) from RNase degradation. However,

this model certainly ignores inherent differences between cellular and extracellular environments that could impact RBP-exRNA interaction. While eCLIP allows us to detect all cellular RBPs-RNA interactions, it is possible that specific subcellular localization or membrane association of this RBP-RNA interaction would not be maintained in the extracellular space. This would notably be the case for a specific RBP-RNA interaction in the nucleus, while only the cytoplasmic form of the RBP would be secreted. Our observation that RBP isoforms with a cytoplasmic localization are preferentially exported in cell CM (e.g., QKI and ILF3) (Figures 3B and S3C) make this scenario highly plausible. In parallel, the variable abundance and nature of RNAses in the different biofluids are likely to play a role in the differential trimming of the protected exRBP-exRNA fragment,[27] or even structured RNA not associated with RBPs, potentially leading to biofluid-specific exRNA sequences (Figure S2C). Finally, we could speculate that some secreted proteins, like extracellular kinases,[28] could potentially modulate exRBP binding activities, leading to release and fast degradation of their protected exRNA fragment while not impacting this interaction in cells. It will therefore be important in the future to develop methods to improve the sensitivity of CLIP experiments to directly access exRBP-exRNA interactions in biofluids and CM.

Our map is not complete for a number of reasons, including potential ascertainment bias. For example, serum and saliva have more detectable RBPs than other biofluids for reasons that may not be of biological significance. We suspect that these two biofluids either have (1) a lower background (e.g., saliva has abundant exogenous RNA and lower levels of endogenous RNA), making fragments that are present easier to detect, or (2) a biological process like coagulation in serum may allow more RBP-bound regions to be available for detection.

The completeness of our map is further limited because of our focus on examining only the first batch of ENCODE eCLIP-profiled RBPs, which represents a small fraction of potential RBPs carrying exRNA. As work with eCLIP profiling expands, the methodology outlined here will allow us to compile a more comprehensive resource illuminating the full spectrum of genes represented by exRNA fragments carried by exRBPs. Based on our data, this may include a majority of human protein-coding genes. This possibility is supported by the fact that 30% of human protein-coding genes include an exRNA fragment that overlaps with loci found by eCLIP to be bound by one of the 150 RBPs considered.

Our resource enables future investigation of exRNA and exRBP biology, including associations with specific carriers that include vesicular, lipoprotein, exomere, and supermere carriers that originate from defined tissues and cell types. RBCs are an obvious "suspect" source cell for many exRBPs and their cargo. Indeed, our results predict the presence of a large number of RBPs in RBC-derived EVs. Moreover, these results suggest that our exRBP data resource, in conjunction with independent profiling experiments (such as RNA-seq performed on RBC EVs), may help precisely identify the cellular source of these exRBPs.

Taken together, the extended exRNA Atlas resource and results of our analyses lay a foundation for the investigation of exRBP-mediated extracellular communication and development of a new class of biomarkers.

### Limitations of the study

Validation of RBPs is not always possible because of technical limitations, such as use of a 30-kDa filter, which leads to loss of small RBPs and protein extraction methods, which can affect recovery of TBRG4. In addition, because computational detection relies on correlations, the method is more sensitive for RBPs that uniquely map to a large number of distinct fragments, each pair of fragments showing potential Pearson correlation, and for RBPs that are present in more samples, thus giving a stronger Pearson correlation signal for any pair of distinct fragments.

### STAR★METHODS

Detailed methods are provided in the online version of this paper and include the following:

- KEY RESOURCES TABLE
- RESOURCE AVAILABILITY
  - Lead contact
  - Materials availability
  - Data and code availability
- EXPERIMENTAL MODEL AND SUBJECT DETAILS
  - Cell lines and cell culture
- METHOD DETAILS
  - Generation of RBP exRNA intersection data
  - Extension of the exRNA Atlas resource
  - Intersection tool and accessibility
  - RBP enrichment analysis in human biofluids
  - Cell media western blots
  - RBP enrichment analysis across exRNA carriers
  - RBPs associated with red blood cell EVs
- QUANTIFICATION AND STATISTICAL ANALYSIS

#### SUPPLEMENTAL INFORMATION

#### ACKNOWLEDGMENTS

We would like to thank Gene Yeo (USCD) for fruitful discussions and advice for integration of ENCODE eCLIP datasets for this study. This publication was supported in part by NIH common fund 1UG3TR002881-01, 1U54DA036134-01, 1U54DA049098-01, 1U54DA049098-01S1, 1UH3TR002881, and OT2OD030547-01S1 (to A.M.) and 5UG3TR002881-02 (to I.G. and J.J.). E.V.N. is a CPRIT Scholar in Cancer Research (RR200040). This work was also supported by NIH common fund 4UH3CA241703-03 (to B.M. and R.L.R.) and a Swiss National Center of Competence in Research (NCCR) in Research RNA & Disease grant (to B.M.). The graphical abstract was created with BioRender.

#### AUTHOR CONTRIBUTIONS

E.L.L., conceptualization, methodology, project administration, software, validation, formal analysis, investigation, data curation, writing, and visualization; A.S., conceptualization, methodology, project administration, software, validation, formal analysis, investigation, writing, and visualization; R.F., conceptualization, methodology, project administration, software, formal analysis, investigation, writing, and visualization; D.C., data curation and software; A.C.S., validation; E.E., formal analysis and investigation; E.A., formal analysis and investigation; A.R.J., data curation, software, and project

administration; I.G., supervision, investigation, and validation; J.R., supervision and visualization; J.C., software and visualization; G.P., investigation and validation; M.B.G., supervision; R.P.A., conceptualization; M.E.R., project administration and funding acquisition; J.L.F., methodology and validation; R.J.C., methodology and validation; R.L.R., methodology and validation; I.M.M., funding acquisition and resources; S.S., validation, project administration, and funding acquisition; A.J.d., supervision, investigation, and validation; L.C.L., investigation and validation; Y.-T.W., investigation and validation; C.-F.T., investigation and validation; T.L., investigation and validation; J.J., supervision, investigation, and validation; K.V.K.-J., conceptualization, methodology, investigation, resources, and supervision; E.V.N., validation and supervision; B.M., conceptualization, methodology, formal analysis, investigation, writing, visualization, supervision, project administration, and funding acquisition; A.M., conceptualization, methodology, formal analysis, investigation, resources, writing, supervision, project administration, and funding acquisition.

### DECLARATION OF INTERESTS

E.V.N. is co-founder, member of the Board of Directors, on the SAB, equity holder, and paid consultant for Eclipse BioInnovations. E.V.N.'s interests have been reviewed and approved by the Baylor College of Medicine in accordance with its conflict of interest policies.

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

## STAR★METHODS

### KEY RESOURCES TABLE

| REAGENT or RESOURCE | SOURCE | IDENTIFIER |
|---|---|---|
| Antibodies | | |
| anti-AGO2 | Sino Biological | Cat#50683-R036; RRID: AB_2860516 |
| anti-AQR (IBP160) | Bethyl Laboratories | Cat#A302-547A; RRID: AB_1998964 |
| anti-BCLAF1 (BTF) | Bethyl Laboratories | Cat#A300-608A; RRID: AB_513581 |
| anti-BUD13 | Bethyl Laboratories | Cat#A303-320A; RRID: AB_10952849 |
| anti-DDX3X | Bethyl Laboratories | Cat#A300-474A; RRID: AB_451009 |
| anti-DDX6 | Abcam | Cat# ab174277; RRID: AB_2928958 |
| anti-DGCR8 | Abcam | Cat#ab191875; RRID: AB_2892625 |
| anti-DROSHA (D28B1) | Cell Signaling Technology | Cat#3364; RRID: AB_2238644 |
| anti-EFTUD2 | Bethyl Laboratories | Cat#A300-957A; RRID: AB_805780 |
| anti-FTO | Abcam | Cat#ab126605; RRID: AB_11127120 |
| anti-GRWD1 | Bethyl Laboratories | Cat#A301-576A; RRID: AB_1040030 |
| anti-ILF3 | Abcam | Cat#ab92355; RRID: AB_2049804 |
| anti-LIN28B | Bethyl Laboratories | Cat#A303-588A; RRID: AB_11125329 |
| anti-NCBP2 | Bethyl Laboratories | Cat#A302-553A; RRID: AB_2034872 |
| anti-PPIG (SRCyp) | Bethyl Laboratories | Cat#A302-075A; RRID: AB_1604291 |
| anti-PRPF8 | Bethyl Laboratories | Cat#A303-921A; RRID: AB_2620270 |
| anti-QKI | Abcam | Cat#ab126742; RRID: AB_11129508 |
| anti-RPS3 | Bethyl Laboratories | Cat#A303-841A; RRID: AB_2620192 |
| anti-SF3A3 | Bethyl Laboratories | Cat#A302-506A; RRID: AB_1966116 |
| anti-SF3B4 | Bethyl Laboratories | Cat#A303-950A; RRID: AB_2620299 |
| anti-SSB | Abcam | Cat#ab75927; RRID: AB_10865926 |
| anti-TBRG4 | Bethyl Laboratories | Cat#A301-392A; RRID: AB_938027 |
| anti-TRA2A | Bethyl Laboratories | Cat#A303-779A; RRID: AB_11218175 |
| anti-TROVE2 | Abcam | Cat#ab207416; RRID: AB_2928959 |
| anti-XPO5 | Bethyl Laboratories | Cat#A303-991A; RRID: AB_2620340 |
| anti-YBX3 (ZONAB) | Bethyl Laboratories | Cat#A303-070A; RRID: AB_10893576 |
| anti-ZNF622 (ZPR9) | Bethyl Laboratories | Cat#A304-075A; RRID: AB_2621324 |
| Goat Anti-Rabbit IgG H&L (HRP) | Abcam | Cat#ab205718; RRID: AB_2819160 |
| Goat Anti-Mouse IgG Antibody, HRP conjugate | Sigma-Aldrich | Cat#12–349; RRID: AB_390192 |
| Anti-rabbit IgG, HRP-linked Antibody | Cell Signaling Technology | Cat#7074; RRID: AB_2099233 |
| Chemicals, peptides, and recombinant proteins | | |
| TRI Reagent | Sigma-Aldrich | Cat#93289 |
| Animal Component-Free Cell Attachment Substrate | STEMCELL Technology | Cat#07130 |
| MesenCult™-ACF Plus 500X Supplement | STEMCELL Technology | Cat#05447 |
| Pro293™ a Serum-free Medium | Lonza | Cat#BEBP12-764Q |
| GlutaMAX™ Supplement | GIBCO | Cat#35050061 |
| Penicillin-Streptomycin (10,000 U/mL) | GIBCO | Cat#15140122 |
| Accutase | GIBCO | Cat#A11105 |
| PBS, pH 7.4 | GIBCO | Cat#10010049 |
| CDM-HD Chemically Defined Medium | FiberCell Systems | Cat#LC5800 |
| DMEM, high glucose, GlutaMAX™ Supplement, pyruvate | GIBCO | Cat#31966021 |

*(Continued on next page)*

*Continued*

| REAGENT or RESOURCE | SOURCE | IDENTIFIER |
|---|---|---|
| DMEM | Corning | Cat#10-017-CV |
| Fetal Bovine Serum | GIBCO | Cat#10270106 |
| Fibronectin bovine plasma | Sigma-Aldrich | Cat#F4759 |
| MEM Non-Essential Amino Acids Solution (100X) | GIBCO | Cat#11140050 |
| Gelatine Solution | MilliporeSigma | Cat#G1393 |
| Critical commercial assays | | |
| 4–15% Mini-PROTEAN™ TGX Stain-Free™ Protein Gels, 15 well, 15 µl | Biorad | Cat#4568086 |
| Trans-Blot Turbo Mini 0.2 µm PVDF Transfer Packs | Biorad | Cat#1704156 |
| Precision Plus Protein™ Dual Color Standard | Biorad | Cat#1610394 |
| EveryBlot Blocking Buffer | Biorad | Cat#12010020 |
| Clarity Western ECL Substrate | Biorad | Cat#1705060 |
| Centricon Plus-70 Centrifugal Filter, 30kDA cutoff | Millipore | Cat#UFC703008 |
| Aurum Affi-Gel Blue Mini Columns | Biorad | Cat#7326708 |
| Amicon® Ultra-15 Centrifugal Filters | Millipore | Cat#UFC9030 |
| Micro BCA™ Protein Assay Kit | ThermoFisher | Cat#23235 |
| NuPAGE™ LDS Sample Buffer (4X) | ThermoFisher | Cat#NP0007 |
| Chemi Blot Blocking Buffer | Azure Biosystems | Cat#AC2148 |
| Radiance Plus | Azure Biosystems | Cat#AC2103 |
| Deposited data | | |
| Analyzed small RNAseq data in multiple exRNA atlas studies | Murillo et al. 2019[2] | www.exrna-atlas.org |
| Intersected ENCODE eCLIP regions | Van Nostrand et al. 2020[13] | https://www.encodeproject.org/encore-matrix/?type=Experiment&status=released&internal_tags=ENCORE |
| exRNA/RBP intersect data | This manuscript - data is publicly available at www.exrna-atlas.org | Accessions listed in Table S3 |
| RBC EV sequences | This manuscript - data is publicly available at www.exrna-atlas.org | EXR-IGHIR1BloodCellEVs-AN |
| Experimental models: Cell lines | | |
| 293T Flp-in T-REX (female) | Thermo Fisher Scientific | Cat#R78007 |
| DiFi cell line (female) | Laboratory of Robert Coffey | Olive et al., 1993[29] |
| Human Bone Marrow Stromal Cells Derived in ACF Medium (female) | STEMCELL Technology | Cat#70071 |
| Software and algorithms | | |
| Samtools 1.3.1 | Li et al., 2009[30] | http://www.htslib.org/ |
| BEDTools 2.17 | Quinlan and Hall, 2010[31] | https://bedtools.readthedocs.io/en/latest/# |
| R v3.5, v4.0 | R Core Team, 2020[32] | https://www.r-project.org/ |
| Image Lab 6.1 | Bio-Rad Laboratories | https://www.bio-rad.com/en-ch/product/image-lab-software?ID=KRE6P5E8Z |
| exRBP fingerprinting pipeline | This manuscript | https://doi.org/10.5281/zenodo.7706896 |
| Stand alone sample intersection tool | This manuscript | https://doi.org/10.5281/zenodo.7706896 |
| Extracellular RNA Atlas | Subramanian et al., 2015[33] | https://exrna-atlas.org |
| Expression Deconvolution (XDec) | Murillo et al., 2019[2] | https://github.com/BRL-BCM/XDec |

## RESOURCE AVAILABILITY

### Lead contact
Further information and requests for resources and reagents should be directed to and will be fulfilled by the lead contact, Dr. Aleksandar Milosavljevic (amilosav@bcm.edu).

### Materials availability
This study did not generate any new unique reagents.

### Data and code availability
- All generated exRNA/RBP intersect data are available via the exRNA atlas: https://www.exrna-atlas.org in the Post Processed Results Files and are publicly available as of the date of publication. Accession numbers are available in the key resources table and listed in Table S3. Instructions on how to access the RBP intersect files in these studies are present here: https://genboree.org/theCommons/projects/exrna-mads/wiki/exRNA%20Atlas#exRBPs. Red blood cell extracellular vesicle sequences are available via the exRNA atlas: https://www.exrna-atlas.org, and accession numbers are listed in the key resources table. Original western blot images will be shared by the lead contact upon request.
- All original code has been deposited at Github and Zenodo: https://doi.org/10.5281/zenodo.7706896 and is publicly available as of the date of publication. The link is provided in the key resources table.
- Any additional information required to reanalyze the data reported in this paper is available from the lead contact upon request.

## EXPERIMENTAL MODEL AND SUBJECT DETAILS

### Cell lines and cell culture
#### DiFi cell model
DiFi cells (Olive et al., 1993,[29] female) were grown in a C2011 FiberCell bioreactor with 20 kDa pore following manufacturer's instructions (FiberCell Systems, New Market, MD) and using FiberCell systems' defined serum free media (CDM-HD). Specifically, the bioreactor was washed overnight with sterile 1X DPBS (Corning, Corning, NY) and then overnight with high glucose DMEM (hgDMEM/Corning). The bioreactor was treated with 0.5 mg of bovine fibronectin (Sigma, St. Louis, MO) in 20 mL of DMEM for 4 h to overnight. The bioreactor was then washed overnight with complete hgDMEM with 10% bovine growth serum, (1% Penicillin-Streptomycin [Pen/Strep, GIBCO, Dublin/Ireland], 1% Glutamine [GIBCO], 1% non-essential amino acids [GIBCO]). The bioreactor was loaded with 1-5x10^8 DiFi cells in complete hgDMEM with 10% serum and allowed to stand for 1 h before circulating complete DMEM with 10% serum. Glucose levels were monitored daily with a glucometer (CESCO bioengineering, Trevose, PA) and when glucose levels were at half of that in starting media, the media bottle was replaced. In subsequent media changes, the bioreactor went from 10% bovine serum to 5% then to 3%, before switching to 10% CDM-HD (DMEM-HD) media.

#### 293T cell model
293T cells (Thermo Fisher Scientific #R78007, female) were cultured for 48 h in CellBIND (Sigma Aldrich, St. Louis, MO, USA) culture dishes using Pro293 medium (Lonza, Basel, Switzerland) supplemented with 1% GlutaMAX Supplement (Life Technologies, Zug, Switzerland) and 1% (v/v) Penicillin-Streptomycin (10,000 U/mL, Life Technologies, Zug, Switzerland) in a CO2 incubator (New Brunswick Galaxy 170 S, Eppendorf, Schönenbuch, Switzerland) at 37°C, 5% CO2.

#### Human mesenchymal stem cell model
hMSCs (STEMCELL Technology #70071, female) were cultured for 48 h in T75 flasks pre-treated with 0.1% gelatin solution as described by the provider (MilliporeSigma, Burlington, MA, USA) using Pro293 medium (Lonza, Basel, Switzerland) supplemented with 1% GlutaMAX Supplement (Life Technologies, Zug, Switzerland), 1% MesenCult-ACF Plus 500X Supplement (STEMCELL Technologies, Vancouver, Canada) and 1% (v/v) Penicillin-Streptomycin (10,000 U/mL, Life Technologies, Zug, Switzerland) in a CO2 incubator (New Brunswick Galaxy 170 S, Eppendorf, Schönenbuch Switzerland) at 37°C, 5% CO2.

## METHOD DETAILS

### Generation of RBP exRNA intersection data
#### Generation of exRNA sample bedGraphs
A custom pipeline (https://doi.org/10.5281/zenodo.7706896) was created to analyze all human fastq files currently present in the atlas. BAM files from the exceRpt pipeline were sorted using Samtools 1.3.1. genomecov -bg was used to generate bedGraphs using BEDtools 2.17. BedGraphs are stored as xz files.

#### ENCODE RBP file preparation
Hg19 BED files containing the two merged replicates were downloaded from ENCODE (https://www.encodeproject.org/metadata/?type=Experiment&status=released&assay_slims=RNA+binding&award.project=ENCODE&assay_title=eCLIP&biosample_ontology.classification%21=tissue&files.file_type=bed+narrowPeak). Files for each cell line were downloaded leading to 223 files (103 HepG2,

120 K562). To generate merged HepG2+K562 files for the 73 RBPs present in both cell line datasets, files were concatenated and then merged using BEDtools mergeBed -s which took strand into account when creating the intersecting regions. In total, 296 unique files (223 cell line specific and 73 merged across cell line) BED files were used to intersect all human files available in the exRNA atlas.

### Generation of exRNA RBP intersection files
exRNA sample data were intersected with the 296 RBP BED files using BEDtools map in a custom shell script (Figure S1A, https://doi.org/10.5281/zenodo.7706896). Files were then aggregated in two ways using BEDtools unionbedg.

(1) RBP centric, where for a given study, all exRNA samples were intersected with a single RBP. In this case, rows are RBP binding sites from the ENCODE data for that single RBP and columns are individual samples.
(2) Sample centric, where for a given exRNA sample, all the RBP intersects for a single sample were combined. In this case, rows represent loci from all RBP BED files. If RBP A binds from chr1:1-10 and RBP B binds from chr1:5-15 three rows would be created (chr1:1–5, chr1:6–10, chr1:11-15) resulting in ~1.4M rows (1,411,176). Columns then represent specific RBPs, so coverage of a single sample can be localized to single or multiple RBPs.

### Gene annotation of exRNA bound to RBPs
gencode.v19.annotation.gtf was downloaded from gencode. The chr, start, end, ENSMBL_ID, type, gene_name, and strand columns were kept to create a mock BED file. BEDTools was then used to intersect this file with the BED file containing all regions bound by RBPs. The loci were then subset to a single RBP's loci and for each gene that overlapped, the biotype (gene, pseudogene, etc.) was recorded. Genes could be recorded as multiple biotypes – i.e. at chr1: 14926–14931 WASH7P has 1 pseudogene and 5 unprocessed pseudogene annotations. In this case, that region was listed as both a pseudogene and an unprocessed pseudogene. Waffle plots were used to visualize the data via ggplot2 (Figure S1F).

### Integrative Genomic Viewer tracing
To show an example of the Integrative Genomic Viewer traces (Figures S2B and S2C), healthy plasma, urine, and saliva samples were used from an exRNA Atlas study (accession: EXR-KJENS1RID1-AN). Additionally, eCLIP data of BUD13 in K562 cell line was used in BED format. This BED file is available publicly through the ENCORE project as mentioned previously.

## Extension of the exRNA Atlas resource
Intersection data can be downloaded via the exceRpt post processed results and core results files for all samples/studies output via the exRNA Atlas website (www.exrna-atlas.org) or API.

- endogenousAlignments_genome_Aligned.bedgraph.intersect_RBP_all_combined.bed.xz (CORE RESULTS)
  - This archive has files that show where a single sample binds across all RBP regions. There will be 296 columns (1 for each RBP if it was sequenced in a single cell line, but if an RBP was sequenced in two cell lines it is actually present 3 times–1 for each cell line and 1 with regions merged from both cell lines). These files will have ~1.4 million rows which represent the intersected RBP regions (Figure S1A, bottom line – 1 file per RBP).
- endogenousAlignments_genome_Aligned_intersect_individual_RBP.tgz (CORE RESULTS)
  - This file shows the number of reads that a specific biosample has in each of the RBP bound regions (Figure S1A, top 2 lines – 1 file per RBP). This folder contains 296 files (1 for each RBP if it was sequenced in a single cell line, but if an RBP was sequenced in two cell lines it is actually present 3 times–1 for each cell line and 1 with regions merged from both cell lines).
- … _intersect_individual_RBP.combined_samples.tgz (POST PROCESSED RESULTS)
  - This folder contains 296 files (1 for each RBP if it was sequenced in a single cell line, but if an RBP was sequenced in two cell lines it is actually present 3 times–1 for each cell line and 1 with regions merged from both cell lines). In this case you will find a matrix where rows are RBPs binding regions and columns represent an individual exRNA sample, showing how many reads fall into each RBP bound region across the entire study. As these are RBP specific files, the rows will represent a single RBPs binding site directly from ENCODE and not the intersected regions (Figure S1A, top 2 lines – 1 file per RBP).

## Intersection tool and accessibility
This tool accepts two files from the user, one which specifies the study and sample name of data the user wishes to use and the second which specifies the regions of interest in BED format (chromosome, start, end). The tool then performs the intersections with the user provided BED and the user selected sample bedGraphs and returns the intersected file matrix. Tool is available at https://doi.org/10.5281/zenodo.7706896.

## RBP enrichment analysis in human biofluids
### Correlation footprinting
Analysis was performed using custom shell and R scripts (available in https://doi.org/10.5281/zenodo.7706896). Sample centric intersect files were used for each sample of interest and columns were filtered to 150 unique RBPs, using the regions from merged files if an RBP was present in both cell lines. Studies were then combined and any rows with no reads were removed. To create sample specific columns, the 150 columns corresponding to each RBP for each sample were summed together.

All regions present in multiple RBPs were removed. For each of the 150 unique RBPs, all pairwise Pearson correlations were determined. For each locus, coverage across samples was determined and a second random locus that did not belong to the RBP being tested was selected by similar coverage. If a locus of the same coverage was not available, a locus of 1 higher coverage was selected (unless there were no other loci with greater coverage where one with 1 less coverage was selected). Next, the Pearson correlation between the randomly selected loci against all RBP loci was determined by two sided Pearson correlation. A Kolmogorov-Smirnov test was used to determine if the distributions of the correlations within the RBP were different from those made against a random RBP for each loci (Figure S1B).

To determine the RBP level of enrichment, the random replacement locus was correlated with the other random replacement loci by two sided Pearson correlation and a Kolmogorov-Smirnov test was performed to generate p values for the random sets. A second Kolmogorov-Smirnov test was used to test the distribution of the p values for each locus of the correlation distribution against the p values for each locus of the random correlations. Only RBPs with p values showing a distribution smaller than random comparisons were considered (Figure S1C).

### Significant RBPs in plasma and cell media

exRBP candidates were selected by performing the above method over 6 different sets of loci in plasma and cell media data - loci present in 30 samples with a coverage of at least 5, 20 samples with a coverage of at least 5, 10 samples with a coverage of at least 5, 30 samples with a coverage of at least 2, 20 samples with a coverage of at least 2, 10 samples with a coverage of at least 2. RBPs that were significantly more correlated (Kolmogorov-Smirnov test Bonferroni adjusted p value <0.05) than random sets of loci in at least one of the 6 sets were considered. Bonferroni adjusted p values and the highest coverage they are correlated at are reported for the 34 RBPs in Figure S1D.

### Identifying enriched RBP functional classes

Manual annotations for RBP functions were taken from Van Nostrand et al.[13] The number of RBPs (from the 150 available in ENCODE) that fell into a functional class were plotted against the functional classes of the 34 RBPs that were significantly connected in both cell media and plasma (Kolmogorov-Smirnov test Bonferroni p value <0.05, Figures S1D and S1E. Those functional classes which fell above the regression line were considered enriched.

### Significant RBPs in 5 major biofluids

exRBP candidates were selected by performing the above method over 6 different sets of loci in plasma, serum, saliva, CSF and urine - loci present in 30 samples with a coverage of at least 5, 20 samples with a coverage of at least 5, 10 samples with a coverage of at least 5, 30 samples with a coverage of at least 2, 20 samples with a coverage of at least 2, 10 samples with a coverage of at least 2. RBPs that were significantly more correlated (Kolmogorov-Smirnov test Bonferroni adjusted p value <0.05) than random sets of loci in at least one of the 6 sets were considered.

### Map of exRBPs in 5 major biofluids

Each biofluid displays those RBPs which were significantly connected via Kolmogorov-Smirnov test (Bonferroni p value <0.05) regardless of loci coverage and samples were clustered using unsupervised clustering in the gplots heatmap.2 function. (Figure S2A).

### RBPs associated with small RNA biotypes

miRNA, snRNA, and snoRNA annotations were taken from ensemble hg38 v104 and were mapped to hg19 using the UCSC table lift service. piRNA annotations came from piRNAdb v1.7.5, Y RNA annotations came from UCSC v39, and tRNA annotations came from tRNA scan hg19. Loci for each RBP were subset down to a given biotype and then the Pearson correlation analysis was performed over the subset of regions. RBPs were considered significantly associated with a given biotype when the Kolmogorov-Smirnov Bonferroni value was < 0.05.

### Off-strand analysis

To determine the rate of off-strand analysis, positive and negative strand bedGraphs were made from 4 glioblastoma exRNA cell line samples and positive and negative strand RBP BED files were also created. Intersects were performed in both a non-strand specific and strand specific manner and extra reads from the non-strand specific intersections were considered off-strand alignments. The percent of reads that were off-strand was then computed for each RBP. Off-strand percentage was calculated for each of the 4 samples and then the average off-strand reads were reported (Table S4).

## Cell media western blots

### DiFi cell model

Conditioned media were prepared largely as previously described.[34] In short, once DiFi culture was established in DMEM-HD (at least two weeks in DMEM-HD), routine harvest of conditioned media was performed by collecting 20 mL per day. Samples were spun at 250xg for 10 min at 4°C, the supernatant was transferred to a new tube and was centrifuged further at 2000xg for 15 min at 4°C. Several collections of filtered conditioned media were pooled and 1 mL of supernatant at this step was concentrated with 30 kDa Amicon columns (GE), followed by removal of albumin with Aurum Affi-Gel columns (Bio-Rad). After this step, samples were again concentrated with 30 kDa Amicon columns (Millipore), lysed in 1X RIPA buffer, and quantified with Micro BCA (ThermoFisher). Collection of DiFi cells from the bioreactor was performed by spinning collected samples at 250xg, then cell pellet was resuspended in 20 mL of PBS and spun again for 1 minute at 250xg. The supernatant was carefully removed, then the pellet was resuspended in 5-10 mL of PBS and 1 mL aliquots were made in microfuge tubes. Tubes were spun for 1 minute at 10,000xg and the supernatants removed. Resulting pellets were immediately frozen at -80°C. For Western blotting, samples were denatured with 0.1M

DTT and 1x NuPAGE LDS buffer (ThermoFisher), ran on 4-12% Novex Bis-Tris gels, and transferred to PVDF membrane with standard wet transfer. Blots were blocked in Chemi Blot Blocking Buffer (Azure biosystems), incubated for either 1 hour or overnight with primary antibody (1:4000 dilution), washed with TBST, incubated for 2 hours with 1:2000 dilution secondary Anti-Rabbit IgG HRP-linked antibody (CST), washed with TBST, and incubated with Radiance Plus (Azure biosystems) chemiluminescent substrate, and imaged on the Azure c300 system (Figure S2A).

*293T and human mesenchymal stem cell models*

Conditioned media (around 50 ml) collected from 293T grown in 5x 10 cm CellBIND dishes (Sigma Alrich, St. Louis, MO, USA) and hMSCs grown in 3x T75 flasks (TPP, Trasadingen, Switzerland) were spun for 5 minutes at 1000xg, and then filtered using a 0.45 μM pore size syringe filter, before being concentrated approximately 100 times using a 30 kDa centricon-70 ultrafiltration cartridge (Millipore) following manufacturer instructions. Four volumes of TRI reagent (Zymo) were added per volume of concentrated conditioned media to lyse and denature the sample. 3 mL of TRI-reagents was directly added to one of the 10 cm culture dishes to lyse and denature matching cell samples (around 10-20 million cells). Both TRI-reagent lysate were incubated at room temperature for 5 minutes, before being snap frozen and stored at -80°C for downstream processing. Protein extraction from hMSCs and 293T cells and corresponding concentrated conditioned media was performed following a modified TRIzol protocol with precipitation of proteins in 4 volumes of acetone. Protein pellets were resuspended in 1xRIPA/1xLaemmli/2-Mercaptoethanol buffer and boiled at 95°C for 5 minutes. Samples were separated in 4–20% Mini-PROTEAN TGX Stain-Free Protein Gels (Biorad). Proteins were transferred onto PVDF membranes (Trans-Blot Turbo Midi 0.2 μm PVDF Transfer Packs, Biorad), blocked with EveryBlot Blocking Buffer (Biorad) for 15 minutes at room temperature and incubated overnight with primary antibodies diluted in blocking buffer at 4°C. Blots were then washed 3 times for 5 minutes with TBST, incubated with either anti-mouse IgG HRP (Sigma-Aldrich) or anti-rabbit IgG HRP (Abcam) for 1 hour at room temperature. Finally, membranes were washed 3 times for 5 minutes with TBST, incubated with Immun-Star HRP reagent (Biorad) for 5 minutes and imaged using the ChemiDoc Imaging System (Biorad). Overall, around 6 million cells and 10 mL of unconcentrated conditioned media were loaded per gel lane, conveying a three time less cell lysate amount compared to the volume of matched conditioned media at 48 hours incubation (Figures 3B and S3C).

*RBP extracellular/intracellular ratio*

293T and hMSCs Western blots bands were analyzed using Image Lab 6.1 software (Bio-Rad Laboratories). Bands were detected using the *Lane and Bands* tool and quantified by subtracting the background noise (taken via the *Adj. Total Band Volume* values). For each RBP, all the adjusted values were normalized by dividing them by the average value of the three cell lysate replicates. Finally, to calculate the extracellular/intracellular ratio for each RBP, the normalized CM band values were divided by the corresponding normalized cell lysate band values. The barplot shows the average of the extracellular/intracellular ratios among replicates and its standard deviation (Figures 3C and S3D).

### RBP enrichment analysis across exRNA carriers

*Deconvolution analysis*

Each of the 150 RBPs were tested for enrichment against each CT. RBP coverage was deconvoluted into CT specific profiles for each study present in the original atlas deconvolution. The estimated coverage for each locus was then divided by the length of each locus to determine the average base pair coverage. To create one profile for each CT, the average base pair coverage for each locus was calculated by averaging each locus across all profiles for a given CT (CT1 has 11 profiles, CT2 has 7, etc.). The average coverage for all CTs was calculated by taking the mean of all estimated base pair coverage. To determine CT enrichment, the average CT coverage was divided by the average total coverage. This resulted in an enrichment measure for each CT for each RBP where the average was 1.

$$CT\ enrichment\ =\ \frac{average\ for\ 1\ CT\left(\frac{\#\ of\ reads}{Length\ of\ loci}\right)}{average\ for\ all\ CT\left(\frac{\#\ of\ reads}{Length\ of\ loci}\right)}$$

The same calculations were performed using reference data generated from serum and plasma processed via density gradient ultracentrifugation.[2] The 80 samples were intersected with all RBPs and then enrichment was calculated for each class (unfractionated (CT3), fractions 1–3 (CT4), 4–7 (CT1), and 9–12(CT2)).

$$CT\ enrichment\ =\ \frac{average\ for\ 1\ class\left(\frac{\#\ of\ reads}{Length\ of\ loci}\right)}{average\ for\ all\ class\left(\frac{\#\ of\ reads}{Length\ of\ loci}\right)}$$

Consistent RBPs were determined if RBPs showed enrichment in both the CT and its correlated fraction (CT1 = fraction 9–12, CT2 = fraction 4–7, CT3 = unfractionated, CT4 = fraction 1–3).

### Heatmap of RBP CT enrichments

All RBP enrichment was visualized as a heatmap where each column was the average base pair expression for a CT and each row was one RBP. The color was determined by the enrichment value – darker colors were higher enrichments. All RBPs were visualized with no cut offs. Unsupervised hierarchical clustering was performed on the rows to determine sets of RBPs which may represent each CT (Figure S4A).

### Upset plots of RBP CT enrichments

To determine cut off points for visualization, the 75th percentile of enrichment of all RBPs was calculated as 1.34. Those with an enrichment of 1.4 were therefore considered enriched. All sets of RBPs where the enrichment was greater than 1.4 were then visualized using upset plots. All 150 RBPs had at least one CT with 1.4 or greater enrichment and no RBP was enriched in more than 3 CTs. The subset of 34 RBPs that showed significant correlation (Kolmogorov-Smirnov test Bonferroni value of the p value <0.05 over) test in both bulk plasma and cell supernatant were also visualized (Figure S4C).

### Percentage of biotypes bound to RBPs in CTs

Analysis was restricted to 10 studies for which CT proportions were determined by deconvolution based on exRNA expression as described in Murillo et al.[2] Stage 2 deconvolution was applied over binding sites where at least one sample had 5 mapped reads. After applying stage 2 deconvolution, only binding sites that had at least 2 estimated within a given CT were considered to be expressed within that CT. To determine the genic regions that overlapped with the expressed binding sites within each biotype, we determined the transcripts whose genomic coordinates overlapped with the coordinates of each RBP binding site (determined using non-strand specific BEDtool intersect function) and considered only those that were found to be expressed within at least one study currently available on the exRNA Atlas. Transcript annotations for lincRNA, mRNA, snoRNA, and snRNA were determined from Gencode v18 transcript type annotations. In the case of piRNA, miRNA, and tRNA, RBP binding sites were overlapped with piRNA annotations from piRNABank, stem-loop annotations from miRNABase version 21, and tRNA annotations from GtRNAdb. Finally, we determined the number of unique base pairs from genes whose transcripts both passed the coverage cutoff and came from genes encoding transcripts that were bound by an RBP and divided by the number of unique base pairs encompassed by all genes encoding transcripts within a specific biotype. Kruskal-Wallis tests were performed to determine if there was an association between the CTs and the fraction of the genome carried by RBPs within a specific biotype. Post-hoc pairwise Wilcoxon rank-sum tests were performed on significant CTs (Kruskal-Wallis test p value <0.05). All pairwise comparisons were adjusted for multiple hypothesis testing using Holm's method (Figure 4C).

### Percentage of biotypes bound to exRBPs

For each biotype we determined the transcripts where at least one sample across all studies currently available on the exRNA atlas had 5 aligned reads as well as the subset of these transcripts whose genomic coordinates overlapped with at least one RBP binding site in a non-strand specific manner. The numerator in this case was the number of transcripts that passed the coverage cutoff within a given biotype and were bound by an RBP and the denominator was the number of genes encoding all transcripts within the specific biotype being analyzed (Figure S4D).

### RBPs associated with red blood cell EVs
### Red blood cell EV isolation

For isolation of RBC EVs, two 1.5 mL eppendorf tubes with a final volume of 200 μL containing approximately the same number of cells were used (cells were counted and normalized in 1X PBS solution). One 1.5 mL eppendorf tube was separated and not treated (parental / control cells). Another 1.5 mL eppendorf tube containing approximately the same number of cells was used to generate EVs. For EV generation and Size Exclusion Chromatography (SEC), 1 μM of ionomycin was added to the PBS 1X solution containing the cells. Cells were incubated for 1 hour at 37°C and centrifuged at 2500xg for 10 minutes. Supernatant containing the vesicles was transferred to another tube and concentrated using Amicon® Ultra-4 column spun at 4000xg for 1 minute. The concentrated volume was subjected to the SEC and then the EVs portions 1-4 were captured and resuspended in 2 mL of 1X PBS. The volume was again concentrated in 200 μL 1X PBS using Amicon® Ultra-4 column spun at 4000xg for 1 minute. This material was used for RNA extraction according to the miRNeasy Micro Kit manufacturer's protocol. cDNA template obtention, miR-AMP reaction and Real Time PCR protocols followed TaqMan® Advanced miRNA Assay manufacturer recommendations. Results were processed with the exceRpt small RNA-seq pipeline.

### Permutation analysis

Four samples of RBC EV small RNA were utilized for the analysis. Bedgraph and RBP intersect files for each sample were generated using the previously described methods. RBPs were filtered for those containing at least 15 loci with non-zero expression in at least 2 samples. The low sample count precludes the correlation approach previously used for RBP detection, and a permutation testing model was used instead. For each RBP associated loci, the pairwise expression differences between samples was measured. The coefficient of variation (CV) $CV = (\sigma / \mu) \times 100$ was then calculated for the expression differences across the RBP associated loci for all pairwise samples. The loci expression data was then randomly shuffled and permutation testing was performed 1000x to produce a distribution of CVs to compare to our original data. CVs below 0.05 were considered significant. As a control cohort, this analysis was repeated with random sets of loci associated with the 99 included RBPs (Figure 4D).

## QUANTIFICATION AND STATISTICAL ANALYSIS

The 150 RBP footprinting analyses were performed using two Kolmogorov-Smirnov tests over (1) all loci within an RBP vs a randomly distributed set of loci with similar coverage to generate a Bonferroni p value for each locus bound by an RBP, followed by (2) testing the distribution of Bonferroni corrected p values vs p values generated by comparing two random sets of loci (Figures S1A–S1C). The samples met the assumptions of the KS test: they were drawn from the same pool, they were mutually independent, and the data is continuous. Methods used for statistical hypothesis testing and exact n numbers are directly stated in the results and STAR Methods sections. All p values were adjusted using the Bonferroni correction and p values < 0.05 were considered significant. All tests are two way unless noted. RBP extracellular/intracellular ratios tested by Western blot were expressed as mean of normalized signal SD (calculation described in STAR Methods section). Experiments were performed in triplicates.

