## [Document S2. Transparent peer review record for LaPlante and Stürchler et al · Cell Genomics]

Title: exRNA-eCLIP intersection analysis reveals a map of extracellular RNA binding proteins and associated RNAs across major human biofluids and carriers.

Authors: Emily L. LaPlante^{*,1}, Alessandra Stürchler^{*,2,3}, Robert Fullem^{1,4}, David Chen¹, Anne C. Starner⁸, Emmanuel Esquivel^{1,4}, Eric Alsop¹⁶, Andrew R. Jackson¹, Ionita Ghiran⁵, Getulio Pereira⁵, Joel Rozowsky⁶, Justin Chang⁶, Mark Gerstein⁶, Roger P. Alexander⁷, Matthew E. Roth¹, Jeffrey Franklin⁹, Robert Coffey⁹, Robert L. Raffai^{10,10+}, Isabelle M. Mansuy³, Stavros Stavrakis², Andrew deMello², Louise C. Laurent¹¹, Yi-Ting Wang¹², Chia-Feng Tsai¹², Tao Liu¹², Jennifer Jones¹³, Kendall Van Keuren-Jensen¹⁶, Eric Van Nostrand^{8,14}, Bogdan Mateescu^{+,2,3,17,19}, and Aleksandar Milosavljevic^{+,1,4,15,17,18,19}

Summary

Initial submission: Received : 8/13/2022

Scientific editor: Laura Zahn

First round of review: Number of reviewers: 2
Revision invited : 9/16/2022
Revision received : 1/3/2023

Second round of review: Number of reviewers: 2
Accepted : 3/24/2023

Data freely available: Yes

Code freely available: Yes

This transparent peer review record is not systematically proofread, type-set, or edited. Special characters, formatting, and equations may fail to render properly. Standard procedural text within the editor's letters has been deleted for the sake of brevity, but all official correspondence specific to the manuscript has been preserved.

Referees' reports, first round of review

Reviewer #1: The primary goal of this manuscript is to generate a map of RBPs that carry in human biofluids RNA cargo. The paper is well-structured and organized overall, with a pleasant flow. It is simple to understand the goal of the study and the methodology employed to analyze the involvement of RBPs in human biofluids RNA cargo. I appreciate how the authors explained the limitations and biases of their experiments in the discussion section.

I do, however, have certain comments that I would like the authors to address. Moreover, the conclusions appear sometime to be a little forced or redundant in certain passages.

Comments:

- 1) The whole study begins with the correlation steps, which consider only regions bound by a single RBP, eliminating those bound by multiple RBPs. I understand why, and the authors discuss this issue in the discussion section. The risk of making this assumption, however, is that it automatically excludes from the study several, and maybe the majority, of regions that are co-bound by multiple RBPs. I'm concerned that this will introduce a biased starting point. Could the authors provide any concrete examples to better defend their decision?
- 2) Why did the authors choose hg19 over hg38 to generate BAM files?
- 3) The authors claim that a selection of 20 RBPs were tested in HEK293 and DiFi cells in two independent laboratories. However, I noted that the authors only showed the RBPs detected in 293T cells in Figure 3B and Figure S2. These data/western blots should be included by the authors. Also, why did you choose DiFi cell line?
- 4) As shown in Figure 3B, some of these RBPs appear to be present in 293 cells but not in 293 conditioned media. Can the authors explain why in more detail?
- 5) To make Figure 2A easier to read, build an interactome profile or venn diagram for RBPs found in biofluids. Because I'm not sure those names are readable, I'd rather see an overlap and then report the names as supplemental table.
- 6) Will it be feasible to include/list the loci used to identify RBPs in Figure S1B? Another alternative is to include a major figure that shows where the 34 RBPs were found based on the footprint analysis.

Reviewer #2: By integrating ENCODE eCLIP data (150 eCLIP RBPs) and human exRNA data (6930 datasets) from a large number of samples, the authors identified and validated exRBP interactions in the plasma, serum, saliva, urine, and other biological fluids. An underlying premise was the high correlation between the level of RNP and exRNA across samples that represent diverse frequency of RBP-exRNA complexes. The bound exRNA transcripts were dominated by small non-coding RNA and fragments of protein-coding transcripts. ExRBP also bind EVs, lipoprotein particles and RNPs. The studies were extended to include specific RNA biotypes bound by exRBPs that are specific to biological fluids.

While the work is largely descriptive, and likely represent only a small fraction of exRBPs interactions, its major impact is the methodology development, which can be expanded and serve and others as a dynamic tool in the field.

Comments

1. A key premise of the work is that "RNA bound inside cells would also be found in the biofluids well represented by exRNA profiles" may ignore differences between the intracellular and extracellular environments that could impact to RBP-exRNA interactions (Fig. 2B-G). Reasons should be more thoroughly mentioned in the Discussion.
2. Were there differences in exRNA reads that interacted with specific RBPs among difference biological

fluids?

3. Could the author validate the ratios of key RBP expression between the medium and cell lysate in another cell type (in addition to 293T), preferably from a highly relevant primary cell type, such as endothelial cells.
 4. Discussion point about biomarkers: do the authors imply that the population of exRNAs bound by a specific RBP could serve as disease biomarker (vs shifts in the RBP-exRNA landscape?)
 5. (minor) The authors should clarify to the non-expert reader that the interaction of RBP with exRNA cargo associates with specific EVs likely occurs within the EV.
 6. (minor) Many items in the long list of references in the second paragraph on page 5 of the PDF seem repetitive.
-

Authors' response to the first round of review

Comments from Reviewer 1

Comment 1: The whole study begins with the correlation steps, which consider only regions bound by a single RBP, eliminating those bound by multiple RBPs. I understand why, and the authors discuss this issue in the discussion section. The risk of making this assumption, however, is that it automatically excludes from the study several, and maybe the majority, of regions that are co-bound by multiple RBPs. I'm concerned that this will introduce a biased starting point. Could the authors provide any concrete examples to better defend their decision?

Response: Indeed, the majority of regions are associated with multiple RBPs (Figure S3C). The last paragraph in "Experimental validation of computationally predicted exRBPs" now discusses the effect of the restriction to unique loci. Briefly, we made the decision to exclude the non-unique loci from correlation footprinting and focus on the loci that are more likely to be RBP-specific expecting to achieve a better experimental validation rate for the RBPs detected. Prompted by reviewer's comment, we have now asked if that is indeed the case by comparing the two methods. Specifically, for correlation footprinting we used either (1) Only loci unique to the RBP or (2) All loci associated with the RBP, even if they associated with multiple RBPs. We focused our analyses on one biofluid (CSF) because of the very long computing time required for all-vs-all correlation analyses for many thousands of loci across all biofluids. The restriction of correlation footprinting to unique loci improved sensitivity, with more RBPs predicted (Figure S2B). Moreover, all 9 RBPs that were predicted by (1) and not predicted by (2) were experimentally validated by Western blotting (Figure S2B).

Comment 2: Why did the authors choose hg19 over hg38 to generate BAM files?

Response: The choice was made in order to avoid reprocessing of thousands of fastq files in the exRNA Atlas. A future version of the Atlas (to be released in late 2023/early 2024) will provide permanent identifiers for specific regions (similarly to ENCODE cis-regulatory cCRE element identifiers in the Screen database) and their begin/end coordinates across reference assemblies, obviating the need for remapping across the multitude of reference assemblies. Because the hg19 coordinates will suffice for obtaining such identifiers (and thus facilitate remapping) we felt that we should not delay our publication by more than a year.

Comment 3: The authors claim that a selection of 20 RBPs were tested in HEK293 and DiFi cells in two independent laboratories. However, I noted that the authors only showed the RBPs detected in 293T cells in Figure 3B and Figure S4. These data/western blots should be included by the authors. Also, why did you choose DiFi cell line?

Response: The 293T cell line results were prioritized for Figure 3 as they had been done in triplicate whereas DiFi had not. The DiFi results can be found in Figure S4A. DiFi cell line was selected by the ERCC

consortium as the key reagent for cross-consortium collaborative protocol harmonization, validation and benchmarking efforts. To support these activities, Coffey Lab of the ERCC established large scale production of DiFi cell culture conditioned media and EVs. For these reasons, DiFi was also utilized for our validation studies.

Comment 4: As shown in Figure 3B, some of these RBPs appear to be present in 293 cells but not in 293 conditioned media. Can the authors explain why in more detail?

Response: There are indeed a number of RBPs that demonstrated this pattern of expression in 293T cell line. Several plausible scenarios can explain this, which are included in the results and discussion section of the manuscript. Most directly relevant to the results of our analysis, is the scenario where the RBP is involved in processing, splicing, or loading of exRNA but is not physically exported from cells along with the exRNA. This would make the RBP detectable in our exRNA footprinting approach despite not being present in biofluid or cell media. Alternatively, rapid protein degradation or various aspects of protein export and processing can explain poor RBP detection in the conditioned media. This is further covered in the fourth and fifth paragraph of the discussion section.

Comment 5: To make Figure 2A easier to read, build an interactome profile or venn diagram for RBPs found in biofluids. Because I'm not sure those names are readable, I'd rather see an overlap and then report the names as supplemental table.

Response: Following the suggestion we have now created an upset plot that better communicates the distribution of RBPs across biofluids. We moved the original Figure 2A panel into the supplement (Figure S2A).

Comment 6: Will it be feasible to include/list the loci used to identify RBPs in Figure S1B? Another alternative is to include a major figure that shows where the 34 RBPs were found based on the footprint analysis.

Response: These tables are available and we will be shared. Because of the large number of tables (several hundred), including them in the current batch of supplemental materials was not feasible. Irrespective of whether the files can be included in the final batch of supplemental materials, the tables will be shared via the exRNA Atlas along with other publicly shared results (<https://exrna-atlas.org/exat/publicAnalyses>). These tables will be made available by the time of manuscript publication.

Comments from Reviewer 2

Comment 1: A key premise of the work is that "RNA bound inside cells would also be found in the biofluids well represented by exRNA profiles" may ignore differences between the intracellular and extracellular environments that could impact to RBP-exRNA interactions (Fig. 2). Reasons should be more thoroughly mentioned in the Discussion.

Response: We agree and have updated the results and discussion sections accordingly. Briefly, our search for "correlation footprints" was indeed guided by intracellular eCLIP results. The hypothesis was that the "correlations footprints" would be observed. While in principle this heuristic may have missed RBP-RNA interactions that only exist in biofluids, a significant number of exRBPs and associated exRNAs were indeed discovered.

Comment 2: Were there differences in exRNA reads that interacted with specific RBPs among different biological fluids?

Response: While the focus on the current paper is on a more coarse level of analysis to establish baseline knowledge about exRBPs, we envision followup studies looking at this more detailed-level analysis. There are indeed many thousands of loci showing diverse patterns of exRBP-associated reads across biofluids. To illustrate this, we have now included IGV traces of two BUD13 eCLIP binding loci that show distinct patterns of exRNA reads across biofluids (Figure S3E-F).

Comment 3: Could the author validate the ratios of key RBP expression between the medium and cell lysate in another cell type (in addition to 293T), preferably from a highly relevant primary cell type, such as endothelial cells.

Response: We strongly agree with this suggestion. We have now included results on primary Mesenchymal Stem Cells, one of the most widely used cell types in EV and exRNA research. Western blots were performed on cell culture conditioned media and compared with 293T cell line results in Figure 3B-C and discussed in the paper.

Comment 4: Discussion point about biomarkers: do the authors imply that the population of exRNAs bound by a specific RBP could serve as disease biomarker (vs shifts in the RBPexRNA landscape?)

Response: We've updated the discussion section to better explain the potential utility of exRBP associated fragments and exRBP-exRNA complexes as biomarkers.

Comment 5: (Minor) The authors should clarify to the non-expert reader that the interaction of RBP with exRNA cargo associates with specific EVs likely occurs within the EV.

Response: We appreciate the comment and have updated the manuscript to improve clarity of that section for non-expert readers.

Comment 6: (Minor) Many items in the long list of references in the second paragraph on page 5 of the PDF seem repetitive.

Response: We appreciate the comment and have updated the manuscript to improve clarity of that section by reducing the references listed

Referees' report, second round of review

Reviewer #1: The authors were able to address all my concerns and the manuscript is ready to go!

Reviewer #2: I thank the authors for addressing my comments.

Authors' response to the second round of review

None